# Intrinsic Motivation for Encouraging Synergistic Behavior

**Rohan Chitnis**[*]        **Shubham Tulsiani**        **Saurabh Gupta**        **Abhinav Gupta**

MIT Computer Science and Artificial Intelligence Laboratory, Facebook Artificial Intelligence Research
`ronuchit@mit.edu, shubhtuls@fb.com, saurabhg@illinois.edu, gabhinav@fb.com`

## Abstract

We study the role of intrinsic motivation as an exploration bias for reinforcement learning in sparse-reward synergistic tasks, which are tasks where multiple agents must work together to achieve a goal they could not individually. Our key idea is that a good guiding principle for intrinsic motivation in synergistic tasks is to take actions which affect the world in ways that would not be achieved if the agents were acting on their own. Thus, we propose to incentivize agents to take (joint) actions whose effects cannot be predicted via a composition of the predicted effect for each individual agent. We study two instantiations of this idea, one based on the true states encountered, and another based on a dynamics model trained concurrently with the policy. While the former is simpler, the latter has the benefit of being analytically differentiable with respect to the action taken. We validate our approach in robotic bimanual manipulation and multi-agent locomotion tasks with sparse rewards; we find that our approach yields more efficient learning than both 1) training with only the sparse reward and 2) using the typical surprise-based formulation of intrinsic motivation, which does not bias toward synergistic behavior. Videos are available on the project webpage: `https://sites.google.com/view/iclr2020-synergistic`.

## 1 Introduction

Consider a multi-agent environment such as a team of robots working together to play soccer. It is critical for a joint policy within such an environment to produce synergistic behavior, allowing multiple agents to work together to achieve a goal which they could not achieve individually. How should agents learn such synergistic behavior efficiently? A naive strategy would be to learn policies jointly and hope that synergistic behavior emerges. However, learning policies from *sparse, binary rewards* is very challenging – exploration is a huge bottleneck when positive reinforcement is infrequent and rare. In sparse-reward multi-agent environments where synergistic behavior is critical, exploration is an even bigger issue due to the much larger action space.

A common approach for handling the exploration bottleneck in reinforcement learning is to shape the reward using *intrinsic motivation*, as was first proposed by Schmidhuber (1991). This has been shown to yield improved performance across a variety of domains, such as robotic control tasks (Oudeyer et al., 2007) and Atari games (Bellemare et al., 2016; Pathak et al., 2017). Typically, intrinsic motivation is formulated as the agent's prediction error regarding some aspects of the world; shaping the reward with such an error term incentivizes the agent to take actions that "surprise it," and is intuitively a useful heuristic for exploration. But is this a good strategy for encouraging synergistic behavior in multi-agent settings? Although synergistic behavior may be difficult to predict, it could be equally difficult to predict the effects of certain single-agent behaviors; this formulation of intrinsic motivation as "surprise" does not specifically favor the emergence of synergy.

In this paper, we study an alternative strategy for employing intrinsic motivation to encourage synergistic behavior in multi-agent tasks. Our method is based on the simple insight that synergistic behavior leads to effects which would not be achieved if the individual agents were acting alone. So,

---

[*]Work done during an internship at Facebook AI Research.

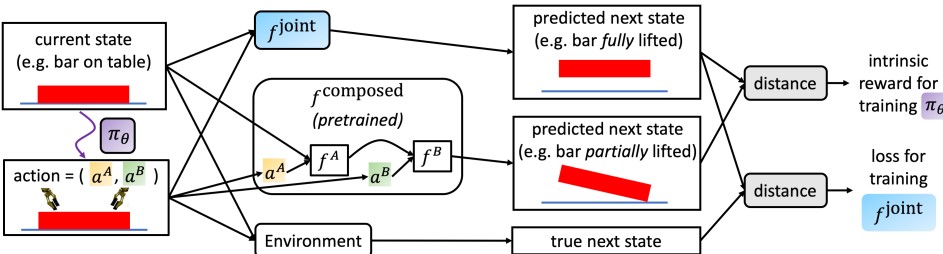

Figure 1: An overview of our approach to incentivizing synergistic behavior via intrinsic motivation. A heavy red bar (requiring two arms to lift) rests on a table, and the policy $\pi_\theta$ suggests for arms $A$ and $B$ to lift the bar from opposite ends. A composition of pretrained single-agent forward models, $f^A$ and $f^B$, predicts the resulting state to be one where the bar is only partially lifted, since neither $f^A$ nor $f^B$ has ever encountered states where the bar is lifted during training. A forward model trained on the complete two-agent environment, $f^{\text{joint}}$, correctly predicts that the bar is fully lifted, very different from the compositional prediction. We train $\pi_\theta$ to prefer actions such as these, as a way to bias toward synergistic behavior. Note that differentiating this intrinsic reward with respect to the action taken does *not* require differentiating through the environment.

we propose to reward agents for joint actions that lead to different results compared to if those same actions were done by the agents individually, in a sequential composition. For instance, consider the task of twisting open a water bottle, which requires two hands (agents): one to hold the base in place, and another to twist the cap. Only holding the base in place would not effect any change in the bottle's pose, while twisting the cap without holding the bottle in place would cause the entire bottle to twist, rather than just the cap. Here, holding with one hand and *subsequently* twisting with the other would not open the bottle, but holding and twisting *concurrently* would.

Based on this intuition, we propose a formulation for intrinsic motivation that leverages the difference between the true effect of an action and the composition of individual-agent predicted effects. We then present a second formulation that instead uses the discrepancy of predictions between a joint and a compositional prediction model. While the latter formulation requires training a forward model alongside learning the control strategy, it has the benefit of being analytically differentiable with respect to the action taken. We later show that this can be leveraged within the policy gradient framework, in order to obtain improved sample complexity over using the policy gradient as-is.

As our experimental point of focus, we study six simulated robotic tasks: four bimanual manipulation (bottle opening, ball pickup, corkscrew rotating, and bar pickup) and two multi-agent locomotion (ant push and soccer). All tasks have sparse rewards: 1 if the goal is achieved and 0 otherwise. These tasks were chosen both because they require synergistic behavior, and because they represent challenging control problems for modern state-of-the-art deep reinforcement learning algorithms (Levine et al., 2016; Lillicrap et al., 2016; Gu et al., 2017; Mnih et al., 2016; Nagabandi et al., 2018). Across all tasks, we find that shaping the reward via our formulation of intrinsic motivation yields more efficient learning than both 1) training with only the sparse reward signal and 2) shaping the reward via the more standard single-agent formulation of intrinsic motivation as "surprise," which does not explicitly encourage synergistic behavior. We view this work as a step toward general-purpose synergistic multi-agent reinforcement learning.

## 2    RELATED WORK

**Prediction error as intrinsic motivation.** The idea of motivating an agent to reach areas of the state space which yield high model prediction error was first proposed by Schmidhuber (1991). Generally, this reward obeys the form $\|f(x) - \hat{f}(x)\|$, i.e. the difference between the predicted and actual value of some function computed on the current state, the taken action, etc. (Barto, 2013; Oudeyer et al., 2007; Bellemare et al., 2016); intrinsic motivation can even be used on its own when no extrinsic reward is provided (Pathak et al., 2017; 2019; Burda et al., 2019; Haber et al., 2018). A separate line of work studies how agents can synthesize a library of *skills* via intrinsic motivation in the absence of extrinsic rewards (Eysenbach et al., 2019). Recent work has also studied the use of surprise-based reward to solve gentle manipulation tasks, with the novel idea of rewarding the agent for errors in its own predictions of the reward function (Huang et al., 2019). In this paper, we will propose formulations of intrinsic motivation that are geared toward multi-agent synergistic tasks.

**Exploration in multi-agent reinforcement learning.** The problem of efficient exploration in multi-agent settings has received significant attention over the years. Lookahead-based exploration (Carmel & Markovitch, 1999) is a classic strategy; it rewards an agent for exploration that reduces its uncertainty about the models of other agents in the environment. More recently, *social motivation* has been proposed as a general principle for guiding exploration (Jaques et al., 2019): agents should prefer actions that most strongly influence the policies of other agents. LOLA (Foerster et al., 2018), though not quite an exploration strategy, follows a similar paradigm: an agent should reason about the impact of its actions on how other agents learn. Our work approaches the problem from a different angle that incentivizes synergy: we reward agents for taking actions to affect the world in ways that would not be achieved if the agents were acting alone.

**Bimanual manipulation.** The field of bimanual, or dual-arm, robotic manipulation has a rich history (Smith et al., 2012) as an interesting problem across several areas, including hardware design, model-based control, and reinforcement learning. Model-based control strategies for this task often draw on hybrid force-position control theory (Raibert et al., 1981), and rely on analytical models of the environment dynamics, usually along with assumptions on how the dynamics can be approximately decomposed into terms corresponding to the two arms (Hsu, 1993; Xi et al., 1996). On the other hand, learning-based strategies for this task often leverage human demonstrations to circumvent the challenge of exploration (Zollner et al., 2004; Gribovskaya & Billard, 2008; Kroemer et al., 2015). In this work, we describe an exploration strategy based on intrinsic motivation.

## 3 APPROACH

Our goal is to enable learning for synergistic tasks in settings with sparse extrinsic rewards. A central hurdle in such scenarios is the exploration bottleneck: there is a large space of possible action sequences that the agents must explore in order to see rewards. In the absence of intermediate extrinsic rewards to guide this exploration, one can instead rely on *intrinsic* rewards that bias the exploratory behavior toward "interesting" actions, a notion which we will formalize.

To accomplish any synergistic task, the agents must work together to affect the environment in ways that would not occur if they were working individually. In Section 3.1, we present a formulation for intrinsic motivation that operationalizes this insight and allows guiding the exploration toward synergistic behavior, consequently learning the desired tasks more efficiently. In Section 3.2, we present a second formulation that is (partially) differentiable, making learning even more efficient by allowing us to compute analytical gradients with respect to the action taken. Finally, in Section 3.3 we show how our formulations can be used to efficiently learn task policies.

**Problem Setup.** Each of the tasks we consider can be formulated as a two-agent finite-horizon MDP (Puterman, 1994).[1] We denote the environment as $\mathcal{E}$, and the agents as $A$ and $B$. We assume a state $s \in \mathcal{S}$ can be partitioned as $s := \langle s^A, s^B, s^{\text{env}} \rangle$, where $s^A \in \mathcal{S}^A$, $s^B \in \mathcal{S}^B$, and $s^{\text{env}} \in \mathcal{S}^{\text{env}}$. Here, $s^A$ and $s^B$ denote the proprioceptive states of the agents, such as joint configurations of robot arms, and $s^{\text{env}}$ captures the remaining aspects of the environment, such as object poses. An action $a \in \mathcal{A}$ is a tuple $a := \langle a^A, a^B \rangle$, where $a^A \in \mathcal{A}^A$ and $a^B \in \mathcal{A}^B$, consisting of each agent's actions.

We focus on settings where the reward function of this MDP is binary and sparse, yielding reward $r^{\text{extrinsic}}(s) = 1$ only when $s$ achieves some desired goal configuration. Learning in such a setup corresponds to acquiring a (parameterized) policy $\pi_\theta$ that maximizes the expected proportion of times that a goal configuration is achieved by following $\pi_\theta$.

Unfortunately, exploration guided only by a sparse reward is challenging; we propose to additionally bias it via an intrinsic reward function. Let $\bar{s} \sim \mathcal{E}(s, a)$ be a next state resulting from executing action $a$ in state $s$. We wish to formulate an intrinsic reward function $r^{\text{intrinsic}}(s, a, \bar{s})$ that encourages synergistic actions and can thereby enable more efficient learning.

### 3.1 COMPOSITIONAL PREDICTION ERROR AS AN INTRINSIC REWARD

We want to encourage actions that affect the environment in ways that would not occur if the agents were acting individually. To formalize this notion, we note that a "synergistic" action is one where

---

[1]Our problem setup and proposed approach can be extended to settings with more than two agents. Details, with accompanying experimental results, are provided in Section 4.5.

the agents acting *together* is crucial to the outcome; so, we should expect a different outcome if the corresponding actions were executed *sequentially*, with each individual agent acting at a time.

Our key insight is that we can leverage this difference between the *true outcome* of an action and the *expected outcome with individual agents acting sequentially* as a reward signal. We can capture the latter via a composition of forward prediction models for the effects of actions by individual agents acting separately. Concretely, let $f^A : \mathcal{S}^{\text{env}} \times \mathcal{S}^A \times \mathcal{A}^A \to \mathcal{S}^{\text{env}}$ (resp. $f^B$) be a single-agent prediction model that regresses to the next environment state resulting from $A$ (resp. $B$) taking an action in isolation.[2] We define our first formulation of intrinsic reward, $r_1^{\text{intrinsic}}(s, a, \bar{s})$, by measuring the prediction error of $\bar{s}^{\text{env}}$ using a composition of these single-agent prediction models:

$$f^{\text{composed}}(s, a) = f^B(f^A(s^{\text{env}}, s^A, a^A), s^B, a^B),$$
$$r_1^{\text{intrinsic}}(s, a, \bar{s}) = \|\bar{s}^{\text{env}} - f^{\text{composed}}(s, a)\|.$$

For synergistic actions $a$, the prediction $f^{\text{composed}}(s, a)$ will likely be quite different from $\bar{s}^{\text{env}}$.

In practice, we pretrain $f^A$ and $f^B$ using data of random interactions in instantiations of the environment $\mathcal{E}$ with only a single active agent. This implies that the agents have already developed an understanding of the effects of acting alone before being placed in multi-agent environments that require synergistic behavior. Note that while random interactions sufficed to learn useful prediction models $f^A$ and $f^B$ in our experiments, this is not essential to the formulation, and one could leverage alternative single-agent exploration strategies to collect interaction samples instead.

### 3.2 PREDICTION DISPARITY AS A DIFFERENTIABLE INTRINSIC REWARD

The reward $r_1^{\text{intrinsic}}(s, a, \bar{s})$ presented above encourages actions that have a synergistic effect. However, note that this "measurement of synergy" for action $a$ in state $s$ requires explicitly observing the outcome $\bar{s}$ of executing $a$ in the environment. In contrast, when humans reason about synergistic tasks such as twisting open a bottle cap while holding the bottle base, we judge whether actions will have a synergistic effect without needing to execute them to make this judgement. Not only is the non-dependence of the intrinsic reward on $\bar{s}$ scientifically interesting, but it is also practically desirable. Specifically, the term $f^{\text{composed}}(s, a)$ is analytically differentiable with respect to $a$ (assuming that one uses differentiable regressors $f^A$ and $f^B$, such as neural networks), but $\bar{s}^{\text{env}}$ is not, since $\bar{s}$ depends on $a$ via the black-box environment. If we can reformulate the intrinsic reward to be analytically differentiable with respect to $a$, we can leverage this for more sample-efficient learning.

To this end, we observe that our formulation rewards actions where the expected outcome under the compositional prediction differs from the outcome when the agents act together. While we used the observed state $\bar{s}$ as the indication of "outcome when the agents act together," we could instead use a *predicted* outcome here. We therefore additionally train a *joint* prediction model $f^{\text{joint}} : \mathcal{S} \times \mathcal{A} \to \mathcal{S}^{\text{env}}$ that, given the states and actions of *both* agents, and the environment state, predicts the next environment state. We then define our second formulation of intrinsic reward, $r_2^{\text{intrinsic}}(s, a, \cdot)$, using the disparity between the predictions of the joint and compositional models:

$$r_2^{\text{intrinsic}}(s, a, \cdot) = \|f^{\text{joint}}(s, a) - f^{\text{composed}}(s, a)\|.$$

Note that there is no dependence on $\bar{s}$. At first, this formulation may seem less efficient than $r_1^{\text{intrinsic}}$, since $f^{\text{joint}}$ can at best only match $\bar{s}^{\text{env}}$, and requires being trained on data. However, we note that this formulation makes the intrinsic reward analytically differentiable with respect to the action $a$ executed; we can leverage this within the learning algorithm to obtain more informative gradient updates, as we discuss further in the next section.

**Relation to Curiosity.** Typical approaches to intrinsic motivation (Stadie et al., 2015; Pathak et al., 2017), which reward an agent for "doing what surprises it," take on the form $r_{\text{non-synergistic}}^{\text{intrinsic}}(s, a, \bar{s}) = \|f^{\text{joint}}(s, a) - \bar{s}^{\text{env}}\|$. These curiosity-based methods will encourage the system to keep finding new behavior that surprises it, and thus can be seen as a technique for curiosity-driven skill discovery. In contrast, we are focused on synergistic multi-agent tasks with an extrinsic (albeit sparse) reward, so our methods for intrinsic motivation are not intended to encourage a diversity of learned behaviors, but rather to bias exploration to enable sample-efficient learning for a given task.

---

[2]As the true environment dynamics are stochastic, it can be useful to consider probabilistic regressors $f$. However, recent successful applications of model-based reinforcement learning (Nagabandi et al., 2018; Clavera et al., 2019) have used deterministic regressors, modeling just the maximum likelihood transitions.

### 3.3 LEARNING SPARSE-REWARD SYNERGISTIC TASKS

We simultaneously learn the joint prediction model $f^{\text{joint}}$ and the task policy $\pi_\theta$. We train $\pi_\theta$ via reinforcement learning to maximize the expected total shaped reward $r^{\text{full}} = r^{\text{intrinsic}}_{i \in \{1,2\}} + \lambda \cdot r^{\text{extrinsic}}$ across an episode. Concurrently, we make dual-purpose use of the transition samples $\{(s, a, \bar{s})\}$ collected during the interactions with the environment to train $f^{\text{joint}}$, by minimizing the loss $\|f^{\text{joint}}(s, a) - \bar{s}^{\text{env}}\|$. This simultaneous training of $f^{\text{joint}}$ and $\pi_\theta$, as was also done by Stadie et al. (2015), obviates the need for collecting additional samples to pretrain $f^{\text{joint}}$ and ensures that the joint prediction model is trained using the "interesting" synergistic actions being explored. Full pseudocode is provided in Appendix A.

Our second intrinsic reward formulation allows us to leverage differentiability with respect to the action taken to make learning via policy gradient methods more efficient. Recall that any policy gradient algorithm (Schulman et al., 2017; 2015; Williams, 1992) performs gradient ascent with respect to policy parameters $\theta$ on the expected reward over trajectories: $J(\theta) := \mathbb{E}_\tau[r^{\text{full}}(\tau)]$. Expanding, we have $J(\theta) = \mathbb{E}_\tau \left[ \sum_{t=0}^{T} r^{\text{full}}(s_t, a_t, \cdot) \right] = \mathbb{E}_\tau \left[ \sum_{t=0}^{T} r^{\text{intrinsic}}_2(s_t, a_t, \cdot) + \lambda \cdot r^{\text{extrinsic}}(s_t) \right]$, where $T$ is the horizon. We show in Appendix B that the gradient can be written as:

$$\nabla_\theta J(\theta) = \sum_{t=0}^{T} \mathbb{E}_{\tau_t}[r^{\text{full}}(s_t, a_t, \cdot)\nabla_\theta \log p_\theta(\bar{\tau}_t)] + \mathbb{E}_{\bar{\tau}_t}[\nabla_\theta \mathbb{E}_{a_t \sim \pi_\theta(s_t)}[r^{\text{intrinsic}}_2(s_t, a_t, \cdot)]]. \quad (1)$$

Here, $\tau_t := \langle s_0, a_0, ..., s_t, a_t \rangle$ denotes a trajectory up to time $t$, and $\bar{\tau}_t := \langle s_0, a_0, ..., s_t \rangle$ denotes the same but *excluding* $a_t$. Given a state $s_t$, and assuming a differentiable way of sampling $a_t \sim \pi_\theta(s_t)$, such as using the reparameterization trick (Kingma & Welling, 2014), we can analytically compute the inner gradient in the second term since $r^{\text{intrinsic}}_2(s_t, a_t, \cdot)$ is differentiable with respect to $a_t$ (again, assuming the regressors $f^A$, $f^B$, and $f^{\text{joint}}$ are differentiable). In Equation 1, the first term is similar to what typical policy gradient algorithms compute, with the difference being the use of $p_\theta(\bar{\tau}_t)$ instead of $p_\theta(\tau_t)$; the intuition is that we should not consider the effects of $a_t$ here since it gets accounted for by the second term. In practice, however, we opt to treat the policy gradient algorithm as a black box, and simply add (estimates of) the gradients given by the second term to the gradients yielded by the black-box algorithm. While this leads to double-counting certain gradients (those of the expected reward at each timestep with respect to the action at that timestep), our preliminary experiments found this to minimally affect training, and make the implementation more convenient as one can leverage an off-the-shelf optimizer like PPO (Schulman et al., 2017).

## 4 EXPERIMENTS

We consider both bimanual manipulation tasks and multi-agent locomotion tasks, all of which require synergistic behavior, as our testbed. We establish the utility of our proposed formulations by comparing to baselines that do not use any intrinsic rewards, or use alternative intrinsic reward formulations. We also consider ablations of our method that help us understand the different intrinsic reward formulations, and the impact of partial differentiability. In Section 4.5, we show that our approach, with minor adaptations, continues to be useful in domains with more than two agents.

### 4.1 EXPERIMENTAL SETUP

We consider four bimanual manipulation tasks: bottle opening, ball pickup, corkscrew rotating, and bar pickup. These environments are suggested as bimanual manipulation tasks by Chitnis et al. (2019). Furthermore, we consider two multi-agent locomotion tasks: ant push (inspired by the domain considered by Nachum et al. (2019)) and soccer (adapted from the implementation provided alongside Liu et al. (2019)). All tasks involve sparse rewards, and require effective use of both agents to be solved. We simulate all tasks in MuJoCo (Todorov et al., 2012). Now, we describe the tasks, state representations, and action spaces.

**Environments.** The four manipulation tasks are set up with 2 Sawyer arms at opposite ends of a table, and an object placed on the table surface. Two of these tasks are visualized in Figure 2, alongside the two multi-agent locomotion tasks.

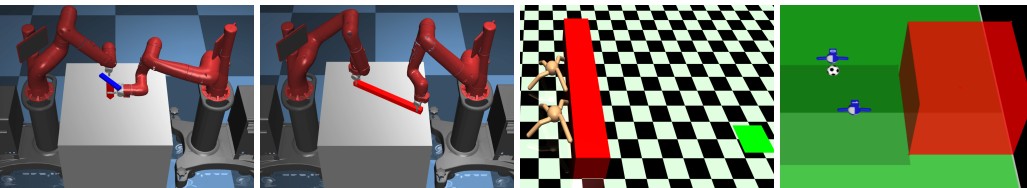

Figure 2: Screenshots of two of our manipulation tasks and our locomotion tasks. From left to right: corkscrew rotating, bar pickup, ant push, soccer. These tasks are all designed to require two agents. We learn policies for these tasks given only sparse binary rewards, by encouraging synergistic behavior via intrinsic motivation.

- *Bottle Opening*: The goal is to rotate a cuboidal bottle cap, relative to a cuboidal bottle base, by $90°$. The bottle is modeled as two cuboids on top of one another, connected via a hinge joint, such that in the absence of opposing torques, both cuboids rotate together. We vary the location and size of the bottle across episodes.
- *Ball Pickup*: The goal is to lift a slippery ball by 25cm. The ball slips out when a single arm tries to lift it. We vary the location and coefficient of friction of the ball across episodes.
- *Corkscrew Rotating*: The goal is to rotate a corkscrew relative to its base by $180°$. The corkscrew is modeled as a handle attached to a base via a hinge joint, such that in the absence of opposing torques, both rotate together. We vary the location and size of the corkscrew across episodes.
- *Bar Pickup*: The goal is to lift a long heavy bar by 25cm. The bar is too heavy to be lifted by a single arm. We vary the location and density of the bar across episodes.
- *Ant Push*: Two ants and a large block are placed in an environment. The goal is for the ants to move the block to a particular region. To control the block precisely, the ants need to push it together, as they will often topple over when trying to push the block by themselves.
- *Soccer*: Two soccer-playing agents and a soccer ball are placed in an environment. The goal is for the ball to be kicked into a particular region, *after* having been in the possession of each agent for any amount of time. Therefore, the agents must both contribute to the movement of the ball.

See Section 4.5 for results on three-agent versions of the Ant Push and Soccer environments.

**State Representation.** The internal state of each agent consists of proprioceptive features: joint positions, joint velocities, and (for manipulation tasks) the end effector pose. The environment state consists of the current timestep, geometry information for the object, and the object pose. We use a simple Euclidean metric over the state space. All forward models predict the change in the object's world frame pose, via an additive offset for the 3D position and a Hamilton product for the orientation quaternion. The orientation is not tracked in the soccer task.

**Action Space.** To facilitate learning within these environments, we provide the system with a discrete library of generic skills, each parameterized by some (learned) continuous parameters. Therefore, our stochastic policy $\pi_\theta$ maps a state to 1) a distribution over skills for agent $A$ to use, 2) a distribution over skills for agent $B$ to use, 3) means and variances of independent Gaussian distributions for every continuous parameter of skills for $A$, and 4) means and variances of independent Gaussian distributions for every continuous parameter of skills for $B$. These skills can either be hand-designed (Wolfe et al., 2010; Srivastava et al., 2014) or learned from demonstration (Kroemer et al., 2015); as this is not the focus of our paper, we opt to simply hand-design them. While executing a skill, if the agents are about to collide with each other, we attempt to bring them back to the states they were in before execution. For manipulation tasks, if we cannot find an inverse kinematics solution for achieving a skill, it is not executed, though it still consumes a timestep. In either of these cases, the reward is 0. See Appendix C for more details on these environments.

## 4.2 IMPLEMENTATION DETAILS

**Network Architecture.** All forward models and the policy are 4-layer fully connected neural networks with 64-unit hidden layers, ReLU activations, and a multi-headed output to capture both the actor and the critic. Bimanual manipulation tasks are built on the Surreal Robotics Suite (Fan et al., 2018). For all tasks, training is parallelized across 50 workers.

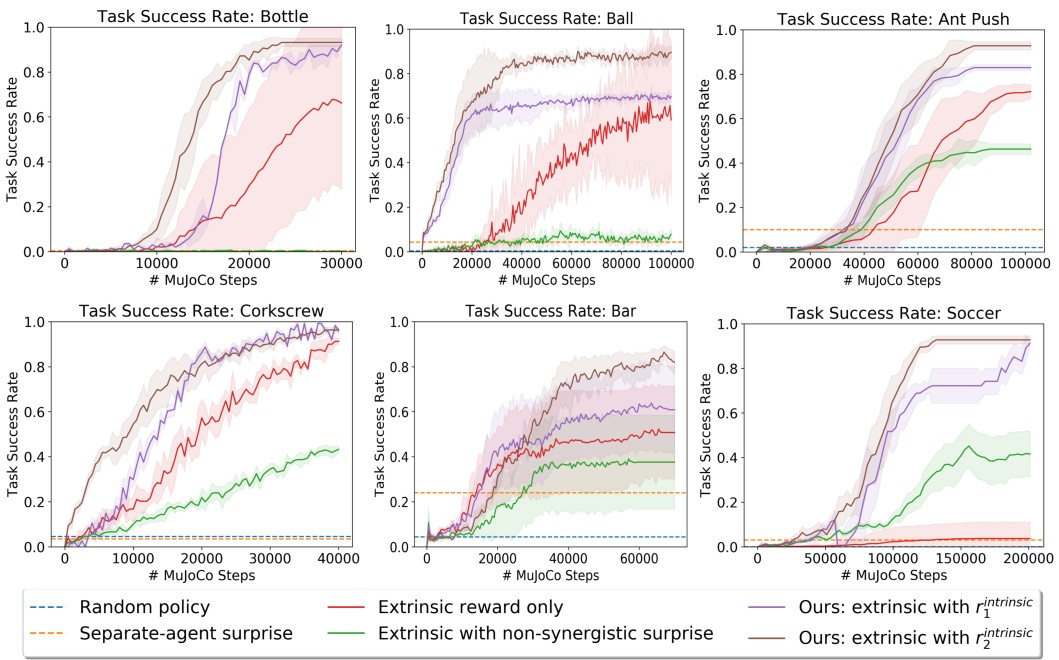

Figure 3: Learning curves for each of our environments. Each curve depicts an average across 5 random seeds, with standard deviations shaded. We see that it is much more sample-efficient to shape the reward via $r_1^{intrinsic}$ or $r_2^{intrinsic}$ than to rely only on the extrinsic, sparse reward signal. Also, typical formulations of intrinsic motivation as surprise do not work well for synergistic tasks because they encourage the system to affect the environment in ways it cannot currently predict, while our approach encourages the system to affect the environment in ways neither agent would if acting on its own, which is a useful bias for learning synergistic behavior.

**Training Details.** Our proposed synergistic intrinsic rewards rely on forward models $f^A$, $f^B$, and $f^{joint}$. We pretrain the single-agent model $f^A$ (resp. $f^B$) on $10^5$ samples of experience with a random policy of only agent $A$ (resp. $B$) acting. Note that this pretraining does not use any extrinsic reward, and therefore the number of steps under the extrinsic reward is comparable across all the approaches. The joint model $f^{joint}$ and policy $\pi_\theta$ start from scratch, and are optimized concurrently. We set the trade-off coefficient $\lambda = 10$ (see Appendix D). We use the stable baselines (Hill et al., 2018) implementation of PPO (Schulman et al., 2017) as our policy gradient algorithm. We use clipping parameter 0.2, entropy loss coefficient 0.01, value loss function coefficient 0.5, gradient clip threshold 0.5, number of steps 10, number of minibatches per update 4, number of optimization epochs per update 4, and Adam (Kingma & Ba, 2015) with learning rate 0.001.

### 4.3 BASELINES

- *Random policy*: We randomly choose a skill and parameterization for each agent, at every step. This baseline serves as a sanity check to ensure that our use of skills does not trivialize the tasks.
- *Separate-agent surprise*: This baseline simultaneously executes two independent single-agent curiosity policies that are pretrained to maximize the "surprise" rewards $\|f^A(s,a) - \bar{s}^{env}\|$ and $\|f^B(s,a) - \bar{s}^{env}\|$ respectively.
- *Extrinsic reward only*: This baseline uses only extrinsic sparse rewards $r^{extrinsic}$, without shaping.
- *Non-synergistic surprise*: We learn a joint two-agent policy to optimize for the extrinsic reward and the joint surprise: $r^{full} = r_{non\text{-}synergistic}^{intrinsic} + \lambda \cdot r^{extrinsic}$. This encourages curiosity-driven skill discovery but does not explicitly encourage synergistic multi-agent behavior.

### 4.4 RESULTS AND DISCUSSION

Figure 3 shows task success rates as a function of the number of interaction samples for the different methods on each environment. We plot average success rate over 5 random seeds using solid lines, and shade standard deviations. Now, we summarize our three key takeaways.

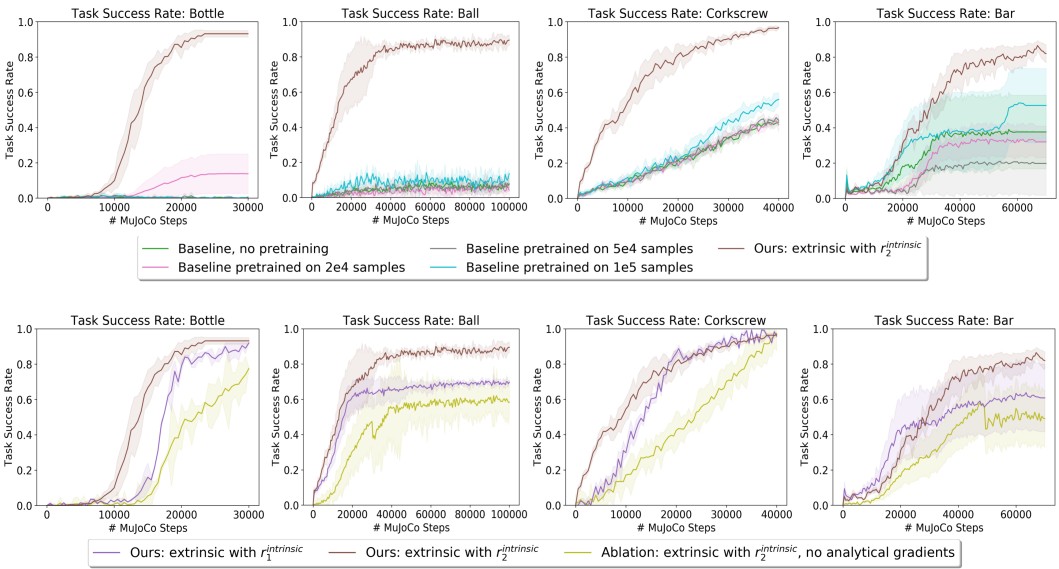

Figure 4: *Top*: *Non-synergistic surprise* baseline with varying amounts of pretraining for the joint model $f^{\text{joint}}$. We see that pretraining this joint model does not yield much improvement in performance, and remains significantly worse than our method (brown curve). This is sensible since the baseline does not explicitly encourage synergistic behavior, as we do. *Bottom*: Ablation showing the impact of using analytical gradients on sample efficiency. $r_2^{\text{intrinsic}}$ only performs better than $r_1^{\text{intrinsic}}$ when leveraging the partial differentiability.

**1) Synergistic intrinsic rewards boost sample efficiency.** The tasks we consider are hard and our use of parameterized skills does not trivialize the tasks. Furthermore, these tasks require coordination among the two agents, and so *Separate-agent surprise* policies do not perform well. Given enough training samples, *Extrinsic reward only* policies start to perform decently well. However, our use of synergistic intrinsic rewards to shape the extrinsic rewards from the environment accelerates learning, solving the task consistently with up to $5\times$ fewer samples in some cases.

**2) Synergistic intrinsic rewards perform better than non-synergistic intrinsic rewards.** Policies that use our synergistic intrinsic rewards also work better than the *Non-synergistic surprise* baseline. This is primarily because the baseline policies learn to exploit the joint model rather than to behave synergistically. This also explains why *Non-synergistic surprise* used together with extrinsic reward hurts task performance (green vs. red curve in Figure 3). Past experiments with such surprise models have largely been limited to games, where progress is correlated with continued exploration (Burda et al., 2019); solving robotic tasks often involves more than just surprise-driven exploration. Figure 4 (top) gives additional results showing that our method's competitive advantage over this baseline persists even if we allow the baseline additional interactions to pretrain the joint prediction model $f^{\text{joint}}$ without using any extrinsic reward (similar to our method's pretraining for $f^{\text{composed}}$).

**3) Analytical gradients boost sample efficiency.** In going from $r_1^{\text{intrinsic}}$ (compositional prediction error) to $r_2^{\text{intrinsic}}$ (prediction disparity), we changed two things: 1) the reward function and 2) how it is optimized (we used Equation 1 to leverage the partial differentiability of $r_2^{\text{intrinsic}}$). We conduct an ablation to disentangle the impact of these two changes. Figure 4 (bottom) presents learning curves for using $r_2^{\text{intrinsic}}$ *without* analytical gradients, situated in comparison to the previously shown results. When we factor out the difference due to optimization and compare $r_1^{\text{intrinsic}}$ and $r_2^{\text{intrinsic}}$ as different intrinsic reward formulations, $r_1^{\text{intrinsic}}$ performs better than $r_2^{\text{intrinsic}}$ (purple vs. yellow curve). This is expected because $r_2^{\text{intrinsic}}$ requires training an extra model $f^{\text{joint}}$ concurrently with the policy, which at best could match the true $\bar{s}^{\text{env}}$. Leveraging the analytical gradients, though, affords $r_2^{\text{intrinsic}}$ more sample-efficient optimization (brown vs. purple curve), making it a better overall choice.

We have also tried using our formulation of intrinsic motivation *without* extrinsic reward ($\lambda = 0$); qualitatively, the agents learn to act synergistically, but in ways that do not solve the "task," which is sensible since the task is unknown to the agents. See the project webpage for videos of these results. Furthermore, in Appendix D we provide a plot of policy performance versus various settings of $\lambda$.

### 4.5 EXTENSION: MORE THAN TWO AGENTS

It is possible to extend our formulation and proposed approach to more than two agents. Without loss of generality, suppose there are three agents $A$, $B$, and $C$. The only major change is in the way that we should compute the compositional prediction: instead of $f^{composed}(s, a) = f^B(f^A(s^{env}, s^A, a^A), s^B, a^B)$, we use $f^{composed}(s, a) = f^C(f^B(f^A(s^{env}, s^A, a^A), s^B, a^B), s^C, a^C)$. One issue is that as the number of agents increases, the ordering of the application of single-agent forward models within $f^{composed}$ becomes increasingly important. To address this, we also tried evaluating $f^{composed}$ as an average across the predictions given by all six possible orderings of application, but we did not find this to make much difference in the results. We leave a thorough treatment of this important question to future work.

We tested this approach on three-agent versions of the ant push and soccer environments, and found that it continues to provide a useful bias. See Figure 5. In three-agent ant push, we give harder goal regions for the ants to push the blocks to than in two-agent ant push; these regions were chosen by hand to make all three ants be required to coordinate to solve these tasks, rather than just two as before. In three-agent soccer, all three agents must have possessed the ball before the goal is scored.

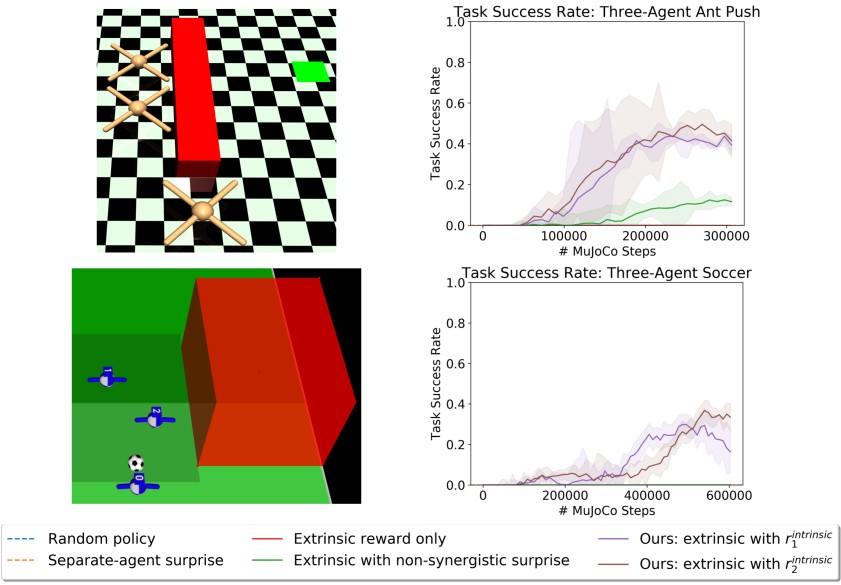

Figure 5: *Left*: Screenshots of three-agent versions of ant push and soccer environments. *Right*: Learning curves for these environments. Each curve depicts an average across 5 random seeds, with standard deviations shaded. In these three-agent environments, taking random actions almost never leads to success due to the exponentially lower likelihood of finding a valid sequence of joint actions leading to the goal, and so using only extrinsic reward does not perform well. It is apparent that our proposed bias toward synergistic behavior is a useful form of intrinsic motivation for guiding exploration in these environments as well.

## 5 CONCLUSION

In this work, we presented a formulation of intrinsic motivation that encourages synergistic behavior, and allows efficiently learning sparse-reward tasks such as bimanual manipulation and multi-agent locomotion. We observed significant benefits compared to non-synergistic forms of intrinsic motivation. Our formulation relied on encouraging actions whose effects would not be achieved by individual agents acting in isolation. It would be beneficial to extend this notion further, and explicitly encourage *action sequences*, not just individual actions, whose effects would not be achieved by individual agents. Furthermore, while our intrinsic reward encouraged synergistic behavior in the single policy being learned, it would be interesting to extend it to learn a diverse set of policies, and thereby discover a broad set of synergistic skills over the course of training. Finally, it would be good to extend the domains to involve more complicated object types, such as asymmetric or deformable ones; especially for deformable objects, engineering better state representations is crucial.

ACKNOWLEDGMENTS

Rohan is supported by an NSF Graduate Research Fellowship. Any opinions, findings, and conclusions expressed in this material are the authors' and need not reflect the views of our sponsors.

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

## A    PSEUDOCODE

Here is full pseudocode of our training algorithm described in Section 3.3:

**Algorithm** TRAIN-SYNERGISTIC-POLICY($\pi_\theta, \mathcal{M}, n, \alpha$)

1    **Input:** $\pi_\theta$, an initial policy.
2    **Input:** $\mathcal{M}$, an MDP for a synergistic task.
3    **Input:** $n$, the number of episodes of data with which to train single-agent models.
4    **Input:** $\alpha$, a step size.
5    **for** $i = 1, 2, ..., n$ **do**
6        Append episode of experience in $\mathcal{M}$ with only agent $A$ acting to data buffer $\mathcal{D}^A$.
7        Append episode of experience in $\mathcal{M}$ with only agent $B$ acting to data buffer $\mathcal{D}^B$.
8    Fit forward models $f^A$, $f^B$ to predict next states in $\mathcal{D}^A$, $\mathcal{D}^B$.     // Pretrained & fixed.
9    $\mathcal{D}^{\text{joint}} \leftarrow \emptyset$        // Data for joint model, only needed if using $r_2^{\texttt{intrinsic}}$.
10   **while** $\pi_\theta$ *has not converged* **do**
11       $\mathcal{D} \leftarrow$ batch of experience tuples $(s_t, a_t, r_t^{\text{extrinsic}}, s_{t+1})$ from running $\pi_\theta$ in $\mathcal{M}$.
12       **if** *using* $r_2^{\text{intrinsic}}$ **then**
13           Append $\mathcal{D}$ to $\mathcal{D}^{\text{joint}}$ and fit forward model $f^{\text{joint}}$ to predict next states in $\mathcal{D}^{\text{joint}}$.
14       **for** $(s_t, a_t, r_t^{\text{extrinsic}}, s_{t+1}) \in \mathcal{D}$ **do**
15           Replace $r_t^{\text{extrinsic}}$ with $r^{\text{full}}(s_t, a_t, s_{t+1})$.     // Shape reward, see Section 3.3.
16       $\nabla_\theta J(\theta) \leftarrow$ POLICYGRADIENT$(\pi_\theta, \mathcal{D})$
17       **if** *using* $r_2^{\text{intrinsic}}$ **then**
18           Update $\nabla_\theta J(\theta)$ with analytical gradients per Equation 1.
19       $\theta \leftarrow \theta + \alpha \nabla_\theta J(\theta)$                    // Or Adam (Kingma & Ba, 2015).

## B    DERIVATION OF EQUATION 1

When using $r_2^{\text{intrinsic}}$, the objective to be optimized can be written as:

$$J(\theta) \equiv \mathbb{E}_\tau[r^{\text{full}}(\tau)] = \mathbb{E}_\tau\left[\sum_{t=0}^{T} r^{\text{full}}(s_t, a_t, \cdot)\right] = \mathbb{E}_\tau\left[\sum_{t=0}^{T} r_2^{\text{intrinsic}}(s_t, a_t, \cdot) + \lambda \cdot r^{\text{extrinsic}}(s_t)\right].$$

We will write $\nabla_\theta J(\theta)$ in a particular way. Let $\bar{\tau}_t = \langle s_0, a_0, s_1, a_1, ..., s_t \rangle$ be a random variable denoting trajectories up to timestep $t$, but *excluding* $a_t$. We have:

$$\nabla_\theta J(\theta) = \nabla_\theta \mathbb{E}_\tau[r^{\text{full}}(\tau)] = \sum_{t=0}^{T} \nabla_\theta \mathbb{E}_{\bar{\tau}_t}[\mathbb{E}_{a_t \sim \pi_\theta(s_t)}[r^{\text{full}}(s_t, a_t, \cdot)]],$$

where we have used the fact that trajectories up to timestep $t$ have no dependence on the future $s_{t+1}, a_{t+1}, ..., s_T$, and we have split up the expectation. Now, observe that the inner expectation, $\mathbb{E}_{a_t \sim \pi_\theta(s_t)}[r^{\text{full}}(s_t, a_t, \cdot)]$, is dependent on $\theta$ since the $a_t$ are sampled from the policy $\pi_\theta$; intuitively, this expression represents the expected reward of $s_t$ with respect to the stochasticity in the current policy. To make this dependence explicit, let us define $r_\theta^{\text{full}}(s_t) := \mathbb{E}_{a_t \sim \pi_\theta(s_t)}[r^{\text{full}}(s_t, a_t, \cdot)]$. Then:

$$\nabla_\theta J(\theta) = \sum_{t=0}^{T} \nabla_\theta \mathbb{E}_{\bar{\tau}_t}[r_\theta^{\text{full}}(s_t)]$$

$$= \sum_{t=0}^{T} \int_{\bar{\tau}_t} \nabla_\theta[p_\theta(\bar{\tau}_t) r_\theta^{\text{full}}(s_t)]\, d\bar{\tau}_t$$

$$= \sum_{t=0}^{T} \int_{\bar{\tau}_t} p_\theta(\bar{\tau}_t) r_\theta^{\text{full}}(s_t) \nabla_\theta \log p_\theta(\bar{\tau}_t) + p_\theta(\bar{\tau}_t) \nabla_\theta r_\theta^{\text{full}}(s_t)\, d\bar{\tau}_t$$

$$= \sum_{t=0}^{T} \mathbb{E}_{\bar{\tau}_t}[r_\theta^{\text{full}}(s_t) \nabla_\theta \log p_\theta(\bar{\tau}_t)] + \mathbb{E}_{\bar{\tau}_t}[\nabla_\theta r_\theta^{\text{full}}(s_t)],$$

where in the second line, we used both the product rule and the REINFORCE trick (Williams, 1992).

Now, let $\tau_t = \langle s_0, a_0, s_1, a_1, ..., s_t, a_t \rangle$ denote trajectories up to timestep $t$, *including $a_t$* (unlike $\bar{\tau}_t$). Putting back $\mathbb{E}_{a_t \sim \pi_\theta(s_t)}[r^{\text{full}}(s_t, a_t, \cdot)]$ in place of $r_\theta^{\text{full}}(s_t)$ gives Equation 1:

$$\nabla_\theta J(\theta) = \sum_{t=0}^{T} \mathbb{E}_{\bar{\tau}_t}[\mathbb{E}_{a_t \sim \pi_\theta(s_t)}[r^{\text{full}}(s_t, a_t, \cdot)]\nabla_\theta \log p_\theta(\bar{\tau}_t)] + \mathbb{E}_{\bar{\tau}_t}[\nabla_\theta \mathbb{E}_{a_t \sim \pi_\theta(s_t)}[r^{\text{full}}(s_t, a_t, \cdot)]]$$

$$= \sum_{t=0}^{T} \mathbb{E}_{\tau_t}[r^{\text{full}}(s_t, a_t, \cdot)\nabla_\theta \log p_\theta(\bar{\tau}_t)] + \mathbb{E}_{\bar{\tau}_t}[\nabla_\theta \mathbb{E}_{a_t \sim \pi_\theta(s_t)}[r_2^{\text{intrinsic}}(s_t, a_t, \cdot)]]. \quad \square$$

In the second line, we have used the facts that $\bar{\tau}_t$ and the extrinsic sparse reward do not depend on $a_t$. Note that we can estimate the term $\mathbb{E}_{\bar{\tau}_t}[\nabla_\theta \mathbb{E}_{a_t \sim \pi_\theta(s_t)}[r_2^{\text{intrinsic}}(s_t, a_t, \cdot)]]$ empirically using a batch of trajectory data $\tau^1, ..., \tau^n$, for any timestep $t$.

## C ADDITIONAL ENVIRONMENT DETAILS

### C.1 MANIPULATION TASKS

We provide additional details about the action space of each manipulation environment.

The following table describes the parameterization of each skill in the library, as well as which environments are allowed to utilize each skill:

| Skill | Environments | Continuous Parameters |
|---|---|---|
| top grasp | bar, ball, bottle | end effector position, end effector z-orientation |
| side grasp | bottle, corkscrew | end effector position, approach angle |
| go-to pose | ball, corkscrew | end effector position, end effector orientation |
| lift | bar, ball | vertical distance to lift end effector |
| twist | bottle | none (wrist joint rotates at current end effector pose) |
| rotate | corkscrew | rotation axis, rotation radius |
| no-op | all | none |

The following table describes the search space of each continuous parameter. Since the object pose is known in simulation, we are able to leverage it in designing these search spaces:

| Continuous Parameter | Environments | Relevant Skills | Search Space |
|---|---|---|---|
| end effector position (unitless) | bar | top grasp | [-1, 1] interpolated position along bar |
| end effector position (meters) | ball, bottle, corkscrew | grasps, go-to pose | [-0.1, 0.1] x/y/z offset from object center |
| end effector z-orientation | bar, ball, bottle | top grasp | $[0, 2\pi]$ |
| approach angle | bottle, corkscrew | side grasp | $[-\frac{\pi}{2}, \frac{\pi}{2}]$ |
| end effector orientation | ball, corkscrew | go-to pose | $[0, 2\pi]$ r/p/y Euler angles converted to quat |
| distance to lift (meters) | bar, bottle | lift | $[0, 0.5]$ |
| rotation axis | corkscrew | rotate | [-0.1, 0.1] x/y offset from object center; vertical |
| rotation radius (meters) | corkscrew | rotate | $[0, 0.2]$ |

Note that our inverse kinematics feasibility checks allow the system to learn to rule out end effector poses which are impossible to reach, since these cause no change in the state other than consuming a timestep, and generate 0 reward.

## C.2   LOCOMOTION TASKS

We provide additional details about the action space of the locomotion environments. For both the ant push and soccer tasks, we follow Nachum et al. (2019) and pre-train four skills: moving up, down, left, and right on the plane. Each skill has one continuous parameter specifying an amount to move. So, at each timestep, the policy must select both which direction to move and how much to move in that direction. All training hyperparameters are unchanged from the manipulation tasks.

## C.3   POLICY ARCHITECTURE

Figure 6 shows a diagram of our policy architecture.

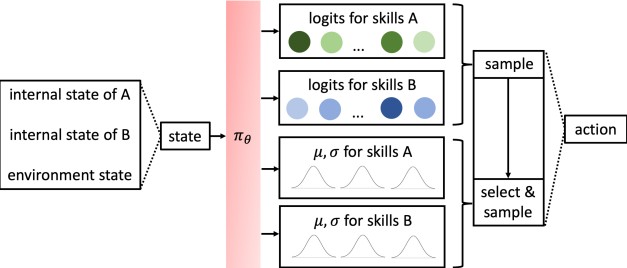

Figure 6: The policy $\pi_\theta$ maps a state to 1) a categorical distribution over skills for $A$, 2) a categorical distribution over skills for $B$, 3) means and variances of independent Gaussian distributions for every continuous parameter of skills for $A$, and 4) means and variances of independent Gaussian distributions for every continuous parameter of skills for $B$. To sample from the policy, we first sample skills for $A$ and $B$, then sample all necessary continuous parameters for the chosen skills from the Gaussian distributions. Altogether, the two skills and two sets of parameters form an action, which can be fed into the forward models for prediction.

## D   IMPACT OF COEFFICIENT $\lambda$

We conducted an experiment to study the impact of the trade-off coefficient $\lambda$ on the performance of the learned policy. When $\lambda = 0$, no extrinsic reward is used, so the agents learn to act synergistically, but in ways that do not solve the "task," which is sensible since the task is unknown to them. Our experiments reported in the main text used $\lambda = 10$. See Figure 7 for the results of this experiment.

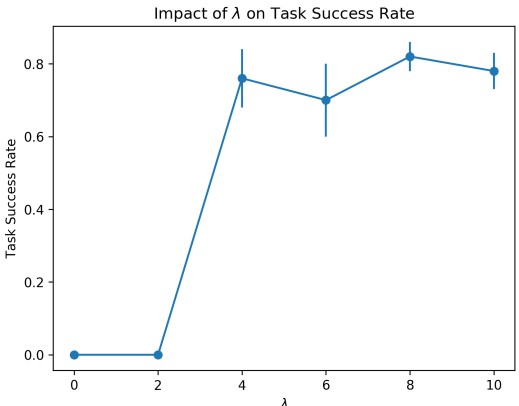

Figure 7: Task success rate of learned policy at convergence for the bar pickup task, using our best-performing reward function $r_2^{\text{intrinsic}}$. Each result is averaged across 5 random seeds, with standard deviations shown. We can infer that once $\lambda$ reaches a high enough value for the extrinsic rewards to outscale the intrinsic rewards when encountered, the agents will be driven toward behavior that yields extrinsic rewards. These extrinsic, sparse rewards are only provided when the task is successfully completed.

