# OpenReview forum: "Intrinsic Motivation for Encouraging Synergistic Behavior"
_ICLR.cc/2020/Conference — Accept (Poster)_

### Official Review · AnonReviewer3 · 2019-10-23
**Official Blind Review #3**

**Rating:** 6

**Review:**

The paper focuses on using intrinsic motivation to improve the exploration process of reinforcement learning agents in tasks with sparse-reward and that require multi-agent to achieve. The authors proposed to encourage the agents toward the actions which changed the world in the ways that "would not be achieved if the agents were acting alone". The experiments are done with dual-arm manipulation.

The idea of guiding the agents toward the actions that they cannot do without concurrent cooperation is interesting. In this paper, it is presented by two types of intrinsic rewards: compositional prediction error and prediction disparity. The core component is the composition of these single-agent prediction model (f^{composed}). Although the formulation proposed is only based on intuition, the authors did enough ablation study to highlight the advantage of this loss function.

Areas to improve:
+ The objects used in the experiment are symmetric, it is good to open your study to the task in which the objects are asymmetric or even deformable.
+ It is good to extend to the problem of multiple-agents (>2), while the order of agents who acts is important to compute "the expected outcome with individual agents acting sequentially" (for now, you only assume that the A will act first then the B acts later).



**Experience Assessment:**

I have read many papers in this area.

**Review Assessment: Checking Correctness Of Derivations And Theory:**

I assessed the sensibility of the derivations and theory.

**Review Assessment: Checking Correctness Of Experiments:**

I assessed the sensibility of the experiments.

**Review Assessment: Thoroughness In Paper Reading:**

I read the paper thoroughly.

---

> ### Author Response · Authors · 2019-11-11
> **Individual Response to Reviewer 3**
>
> Thank you for taking the time to review!
>
> “The objects used in the experiment are symmetric, it is good to open your study to the task in which the objects are asymmetric or even deformable.”
> - We thank the reviewer for this suggestion, which would make a very interesting direction for future work. We have updated the PDF to mention this setting at the end. Also, speaking to a more general point about symmetry, we would like to note that in our bimanual manipulation tasks such as opening a water bottle, the behaviors we learn are not symmetric across the agents: for instance, one agent learns to hold the bottle in place while the other learns to twist the cap.
>
> “extend to the problem of multiple-agents (>2), while the order of agents who acts is important to compute”
> - As highlighted above, we have conducted experiments on a new domain, Ant Push. We have also explored N=3 agents in this setting. Furthermore, as discussed in Appendix E, we have made an initial attempt at addressing the ordering question by computing composition as an average over all possible orderings. Please see the global comment for more details.

---

### Official Review · AnonReviewer1 · 2019-10-24
**Official Blind Review #1**

**Rating:** 8

**Review:**

The paper is technically sound and easy to read. I very much welcome that.

The authors address the important issue of exploration in reinforcement learning. In this case, they propose to use reward shaping to encourage joint-actions whose outcomes deviate from the sequential counterpart. Although the proposed intrinsic reward is targeted at a particular family of two-agent robotic tasks, one can imagine generalizing some of the ideas here to other multi-agent learning tasks.

The authors did mention testing with lambda=0 (intrinsic only), which did not aim to solve the 'task' but would be informative in terms of understanding the results of this particular bias. It would be interesting to include a plot on the effect of lambda across a range, say from 0 to 10.


**Experience Assessment:**

I have published in this field for several years.

**Review Assessment: Checking Correctness Of Derivations And Theory:**

I assessed the sensibility of the derivations and theory.

**Review Assessment: Checking Correctness Of Experiments:**

I carefully checked the experiments.

**Review Assessment: Thoroughness In Paper Reading:**

I read the paper thoroughly.

---

> ### Author Response · Authors · 2019-11-11
> **Individual Response to Reviewer 1**
>
> Thank you for taking the time to review!
>
> “one can imagine generalizing some of the ideas here to other multi-agent learning tasks.”
> - As highlighted above, we have conducted experiments on a new domain, Ant Push. We have also explored N=3 agents in this setting. Please see the global comment for more details.
>
> “The authors did mention testing with lambda=0 (intrinsic only), which did not aim to solve the 'task' but would be informative in terms of understanding the results of this particular bias. It would be interesting to include a plot on the effect of lambda across a range, say from 0 to 10.”
> - We have conducted the requested experiment. A plot and accompanying discussion can be found in Appendix D of the updated PDF.

---

### Official Review · AnonReviewer2 · 2019-10-26
**Official Blind Review #2**

**Rating:** 6

**Review:**

The paper proposes a novel algorithm for encouraging synergistic behavior in multi-agent setups with an intrinsic reward that promotes the agents to work together to achieve states that they cannot achieve individually without cooperation. The paper focuses on a two-agent environment where an approximate forward dynamics model is learnt for each agent, and can be composed sequentially to predict the next environment state given each agent’s action. However, this prediction will be inaccurate if the agent’s affected the environment state in such a way that individual dynamics model cannot predict i.e. synergistic behavior was produced. This prediction error is used as extrinsic reward by the proposed approach, while also having a variant where the true next state is replaced by another approximation of a joint forward model which allows for differentiability of actions with respect to the intrinsic reward. Empirical analysis shows that this intrinsic reward promotes synergetic behavior on two-agent robotic manipulation tasks and achieves better performance that baselines and ablations.

I vote for weak accept as the paper proposes a novel intrinsic reward for promoting synergetic behavior in multi-agent systems, while also demonstrating that such an intrinsic reward can be differentiable if a joint forward dynamics model is approximated in addition to individual forward dynamics models given each agent’s actions. The paper does not show experiments beyond 2 agents and the four robotic manipulation tasks have been shown to work when provided with generic skills as an action space, which requires hand-defining or learning by demonstration.

From the description of the random policy, it is stated that a random policy over skills serves as a sanity check to ensure that the skills do not trivialize the task. This seems to suggest that the extrinsic reward only baseline did not use these skills and was disadvantaged. Some clarification is required here - did the extrinsic reward only baseline use the same skills as the proposed method? If it did, it would obviate the need to have a random policy sanity check.

The paper suggests a baseline for separate arm surprise. In a similar vein, why wasn’t a joint-arm surprise baseline employed, which can basically treat both arms as a single agent?

Synergistic behavior seems hard to achieve for a large number of agents, but the paper does not give insights into whether such an algorithm will work for more than 2 agents. Typically multi-agent systems in prior work have worked with a large number of agents in environments other than robotic manipulation - such experiments may help in strengthening the proposed method.


**Experience Assessment:**

I have read many papers in this area.

**Review Assessment: Checking Correctness Of Derivations And Theory:**

I carefully checked the derivations and theory.

**Review Assessment: Checking Correctness Of Experiments:**

I carefully checked the experiments.

**Review Assessment: Thoroughness In Paper Reading:**

I read the paper at least twice and used my best judgement in assessing the paper.

---

> ### Author Response · Authors · 2019-11-11
> **Individual Response to Reviewer 2**
>
> Thank you for taking the time to review!
>
> “The paper does not show experiments beyond 2 agents” and
> “Synergistic behavior seems hard to achieve for a large number of agents, but the paper does not give insights into whether such an algorithm will work for more than 2 agents. “
> - As highlighted above, we have conducted experiments on a new domain, Ant Push. We have also explored N=3 agents in this setting. Please see the global comment for more details.
>
> “From the description of the random policy, it is stated that a random policy over skills serves as a sanity check to ensure that the skills do not trivialize the task. This seems to suggest that the extrinsic reward only baseline did not use these skills and was disadvantaged. Some clarification is required here - did the extrinsic reward only baseline use the same skills as the proposed method? If it did, it would obviate the need to have a random policy sanity check.”
> - The “extrinsic reward only” baseline does indeed use the same skills as our proposed method. We agree that this obviates the need to have a random policy as a “baseline”; but it is still useful in providing a general sense of how hard the task is.
>
> “The paper suggests a baseline for separate arm surprise. In a similar vein, why wasn’t a joint-arm surprise baseline employed, which can basically treat both arms as a single agent?”
> - Joint-arm surprise in combination with extrinsic reward can be found as the fourth baseline (non-synergistic surprise), introduced in Section 4.3. If the reviewer is instead asking about joint-arm surprise without extrinsic reward, then we hope that the ablation curves in Figure 4 (top row) help address this question. There, we can see that across varying amounts of pre-training the joint-arm surprise baseline without extrinsic reward, the performance of the policy at the start (i.e., x=0) is always low, thereby indicating that using only joint-arm surprise does not lead to solving the task.

---

### Author Response · Authors · 2019-11-11
**Global Response to all Reviewers**

We would like to start by thanking all the reviewers for taking the time to read and provide comments on our work. We are glad that you found the paper interesting and easy to read. We are happy to report that we have conducted experiments on a new domain (Ant Push) and also show results in this domain for N=3 agents. In this global post and in individual replies below, we respond in detail to all the reviewers’ comments/questions.

All reviewers mentioned that it would be interesting to consider extending the work beyond two-agent robotic manipulation. To address this, we developed a new domain, Ant Push, loosely inspired by an experiment in “Multi-Agent Manipulation via Locomotion using Hierarchical Sim2Real.”[1] The results for the two-agent version of this domain can be found in the Experiments section of the updated PDF (Figure 3, rightmost graph), and additional details are in Appendix C. We find that our proposed bias toward synergistic behavior is a useful form of intrinsic motivation for guiding exploration in this environment as well.

We also experiment with a three-agent version of this Ant Push domain, and find that our approach continues to perform well (Appendix E in updated PDF). Note that our approach extends in the natural way; the only major change is the computation of the composed prediction. It remains to be studied what the upper limit of this N is before performance starts to deteriorate.

In response to R3: “It is good to extend to the problem of multiple-agents (>2), while the order of agents who acts is important to compute ‘the expected outcome with individual agents acting sequentially’ (for now, you only assume that the A will act first then the B acts later).”
- When conducting these N=3 experiments, we also tried instead computing the composition by taking the average over all 6 possible orderings of the three forward models. We did not find much difference in the results for our experiments. Nevertheless, we agree that finding the right ordering can be very important as N increases. We leave further investigation of this question to future work.

The N=3 results can be found in Appendix E of the updated PDF.

[1] Nachum, Ofir, et al. "Multi-agent manipulation via locomotion using hierarchical sim2real." arXiv preprint arXiv:1908.05224 (2019).

---

### Decision · Program_Chairs · 2019-12-19

**Decision:**

Accept (Poster)

**Comment:**

The authors address the important issue of exploration in reinforcement learning. In this case, they propose to use reward shaping to encourage joint-actions whose outcomes deviate from the sequential counterpart. Although the proposed intrinsic reward is targeted at a particular family of two-agent robotic tasks, one can imagine generalizing some of the ideas here to other multi-agent learning tasks.

The reviewers agree that the paper is of interest to the ICLR audience.